# Effect of Verbal Encouragement on Postural Balance in Individuals with Intellectual Disabilities

**DOI:** 10.3390/healthcare12100995

**Published:** 2024-05-12

**Authors:** Ghada Jouira, Dan Iulian Alexe, Cristina Ioana Alexe, Haithem Rebai, Alina Ionela Cucui, Ana-Maria Vulpe, Gheorghe Gabriel Cucui, Sonia Sahli

**Affiliations:** 1Research Laboratory Education, Motricité, Sport et Santé (EM2S) LR19JS01, High Institute of Sport and Physical Education of Sfax, University of Sfax, Sfax 3029, Tunisia; jouiraghada0825@gmail.com (G.J.); sonia.sahli@isseps.usf.tn (S.S.); 2Department of Physical and Occupational Therapy, “Vasile Alecsandri” University of Bacău, 600115 Bacău, Romania; 3Department of Physical Education and Sports Performance, “Vasile Alecsandri” University of Bacău, 600115 Bacău, Romania; alexe.cristina@ub.ro (C.I.A.); zaharia.ana@ub.ro (A.-M.V.); 4Tunisian Research Laboratory ‘Sports Performance Optimization’ (LR09SEP01), National Center of Medicine and Science in Sports (CNMSS), Tunis 1002, Tunisia; haithem.rebai@yahoo.fr; 5Department of Physical Education and Sports, “Valahia” University of Târgoviște, 130004 Târgoviște, Romania; ghegabriel.cucui@valahia.ro

**Keywords:** verbal encouragement, intellectual disability, posturography, Y balance test, expanded timed up-and-go test

## Abstract

This study investigated the effect of verbal encouragement (VE) on static and dynamic balance in individuals with intellectual disabilities (IDs). A total of 13 mild IDs and 12 moderate IDs participants underwent static balance tests (bipedal stance on firm surface, under open eyes (OEs) and closed eyes (CEs), and foam surface, unipedal stance on firm surface) and dynamic balance assessments (Y Balance Test (YBT) and Expanded Timed Up-and-Go Test (ETUGT)) under VE and no VE (NO/VE) conditions. VE significantly reduced center of pressure mean velocity (CoP_Vm_) values for mild IDs in firm bipedal CEs conditions. The mild IDs group exhibited improved YBT scores and enhanced ETUGT performances for both groups under VE. Incorporating VE as a motivational strategy in balance training interventions can positively impact static and dynamic balance in individuals with mild IDs, especially in challenging conditions like unipedal stances on firm surfaces.

## 1. Introduction

Maintaining postural balance is an important aspect of human mobility, as it allows individuals to perform daily tasks with stability and confidence, thus enhancing their overall well-being and functional independence. Postural balance encompasses the ability to regulate and stabilize body position during both static and dynamic tasks [1]. Static postural balance activities require an individual’s ability to maintain a stable base of support throughout the assessment, ensuring that postural stability is maintained [2]. Conversely, dynamic postural balance involves the retention of postural balance and orientation of the center of mass over the base of support while body components are in motion [3]. Both static and dynamic postural balance are essential for the successful execution of daily activities [4]. Impairments in postural balance have been associated with an increased risk of falls and are of particular concern for individuals with intellectual disabilities (IDs) [5].

An ID is a lifelong neurodevelopmental disorder characterized by significant deficiencies in both intellectual functioning and adaptive functioning [6]. A diagnosis of an ID is usually determined by an intellectual quotient (IQ) of less than 70, according to the classification provided by the American Psychiatric Association [7]. This classification also classifies severity levels into mild, moderate, severe, and profound IDs, providing a framework for assessing and understanding the extent of cognitive impairments. Strong evidence suggests that individuals with IDs have poor postural balance compared to those without IDs [8,9,10]. Consequently, this population faces an inherent risk of falls and associated injuries [11]. Notably, falls pose a significant concern in this population, resulting in injuries [11,12] and reduced mobility [12]. 

The concept of psychomotricity assumes a significant role, acting as a bridge between the psychological and motor domains [13]. Individuals with IDs frequently manifest distinctive psychological traits, such as a fear of falling, leading to negative emotions such as anxiety and reduced self-confidence [12,14]. In addition, cognitive impairments with low attention and working memory contribute to the fear of falling, further restricting their engagement in activities, limiting their independence, and adversely affecting their overall quality of life [14]. Increased fear is often synonymous with a lack of confidence and motivation in performing motor tasks [15,16]. In this context, individuals with IDs may experience motivation deficits [17], which may influence their willingness to engage in physical activities, and, consequently, affect their motor performance and postural balance [18]. Addressing motivation deficits and promoting positive self-confidence is essential in supporting individuals with IDs in improving their motor tasks and postural balance. Therefore, extrinsic factors such as verbal encouragement (VE) have shown promise in improving confidence, reducing the fear of movement, and thereby enhancing motor performance [19].

VE provides essential external cues that support individuals during motor tasks, enhancing their focus, determination, and confidence across various activities [20,21]. Indeed, scientific evidence, in typical development individuals, highlights that VE has the potential to enhance motivation and motor performance across various activities [22,23,24,25,26]. VE serves as both a cognitive and motivational process reflected in brain activity patterns [27]. Moreover, it acts as a specific external cue, contributing to the efficient establishment of functional connections across brain networks, particularly within the cerebro-cerebellar network that supports skilled movement [28], especially in tasks related to postural balance [29]. Furthermore, several studies showed that using VE during a task can improve a person’s concentration, self-belief, and motivation [21,30,31]. These psychological enhancements are directly connected to performing better in motor tasks [19,28,32]. In fact, VE acts as an additional cognitive and motivational resource, helping individuals overcome mental challenges during motor tasks [24], including balance tasks. Moreover, by making a person more attentive, VE not only helps to focus better [30,31] but also establishes a stronger mind–body connection, leading to improved motor performance [28]. In this context, recent research has even demonstrated that VE improves dynamic balance in individuals with chronic ankle instability [19]. This finding suggests that VE may not only enhance dynamic balance but also enhance task focus and activate higher brain centers, helping individuals with chronic ankle instability overcome fear avoidance, which is a significant factor contributing to balance impairments in this population. Building on these insights, the regular utilization of VE during rehabilitation underscores its potential to motivate individuals with IDs to push their balance limits, enabling them to attain their maximum potential in a safe and controlled environment.

Understanding the effects of VE on static and dynamic postural balance in individuals with IDs holds significant implications. Incorporating VE into balance assessments offers a promising opportunity to improve accessibility and motivation during the evaluation process. By integrating VE into assessments, we create an environment where individuals with IDs receive verbal cues, boosting their confidence and self-assurance inspiring them to surpass their perceived limits. This psychological impact can help reduce anxiety, making assessments more approachable. It can also instill enjoyment and engagement, fundamentally transforming the assessment experience. By understanding how VE influences postural balance, medical staff can tailor interventions to enhance the overall well-being of individuals with IDs. Furthermore, researchers can further refine assessment protocols, considering the positive effects of VE, thereby contributing to the advancement of rehabilitation and therapeutic approaches. As well, caregivers, who play a crucial role in the lives of individuals with IDs, can leverage VE techniques learned from assessments to provide better support and encouragement in daily activities. It is important to consider that the effect of VE on postural balance may differ based on the level of ID. In fact, individuals with different levels of ID may exhibit varying degrees of cognitive impairments, motor deficits, and fear of movement. By examining the differential effects of VE on static and dynamic postural balance across IDs, we can gain knowledge about the individualized needs of this population and tailor interventions accordingly. Therefore, this study aims to investigate the effects of VE on static and dynamic postural balance in individuals presenting with varying levels of IDs. Our hypothesis posits that VE will exert a beneficial influence on static and dynamic postural balance in individuals with IDs. Furthermore, we anticipate that the effect of VE on postural balance may exhibit variability that is dependent on ID severity.

## 2. Materials and Methods

### 2.1. Participants

The sample size was a priori calculated using the software G*power for Windows (version 3.1.9.2; Heinrich Heine University Düsseldorf, North Rhine-Westphalia, Germany) [33]. Considering the lack of published data in the literature concerning the effect of VE on postural balance outcomes in individuals with different levels of IDs, a priori sample size estimation was calculated using a large effect size (Cohen’s f = 0.4). Values for alpha, power, correlation among repeated measures, and the no sphericity correction (ε) were set at 0.05, 0.95, 0.5, and 1, respectively. In total, to reach the desired power, data from at least 16 participants were deemed to be sufficient to minimize the risk of a Type II statistical error.

This study focused on adults with IDs who were attending a special educational center. Prior to data collection, official permissions were obtained from the center authorities, and participants’ personal information and medical records were collected. Informed consent from parents and caregivers was obtained in writing, which included a clear description of the study’s purpose, procedures, and potential risks and benefits. Furthermore, due to their IDs, participants provided verbal consent during this process. A total of 30 adults with mild to moderate IDs (ranging from 20 to 34 years old) met the eligibility criteria and were included. This selection was based on the understanding that individuals with mild to moderate IDs typically do not experience severe physical impairments and can independently take care of themselves. The inclusion criteria for this study were as follows: adults having an ID with an intelligence quotient (IQ) score ranging from 40 to 70 (using the Wechsler Adult Intelligence Scale-Fourth Edition (WAIS–IV) test [34], not using neuroleptic medications or other substances that could affect balance, no history of lower limb injuries or surgeries within the past year, not using assistive devices for balance and walking, and no visual and/or vestibular disorders. Five adults were excluded from the sample based on the inclusion criteria and the results of the developmental coordination disorder list. Ultimately, 13 adults with mild IDs and 12 adults with moderate IDs participated in this study. The independent t-test showed no significant difference in age, height, or weight between the two groups. A significant difference (*p* < 0.001) was observed in IQ scores between the two groups (Table 1). The present study was conducted according to the Declaration of Helsinki, and the protocol was fully approved by the local ethics committee.

### 2.2. Study Design

This study is a crossover comparative design. Two groups of individuals with IDs, 13 with mild IDs and 12 with moderate IDs, were included. Each participant underwent both VE and no VE (NO/VE) conditions, serving as their own control. This study assessed static postural balance using a stabilometric platform in four conditions: firm bipedal with open eyes (OEs), firm bipedal with closed eyes (CEs), firm unipedal, and foam bipedal. Dynamic postural balance was assessed using the Y Balance Test (YBT) and Expanded Timed Up-and-Go Test (ETUGT). Each participant underwent the postural balance assessments twice: once with no VE and then with VE. The same VE was consistently provided to all participants during the VE condition. Participants visited the laboratory twice, with the first visit consisting of a familiarization session three days before the experimental protocol to ensure an understanding of the test procedures, followed by a testing session during the second visit to evaluate the effect of VE on postural balance. We incorporated a 35–45 min rest period between VE and no VE assessments during the same testing session. This extended rest aimed to mitigate potential fatigue effects on postural balance assessments. A standardized assessment order was followed by all participants to maintain consistency and control for order effects.

### 2.3. Measurements

#### 2.3.1. Static Balance

Participants were asked to maintain their static postural balance barefoot as much as possible on a static stabilometric platform (PostureWin© Techno Concept^®^, Cereste, France; 40 Hz frequency, 12-bits A/D conversion) in two postural conditions. In the bipedal stance condition, the participants’ legs were straight, and the (30°) configurations were suggested as feet position, with the heels slightly apart (3 to 5 cm) (French Posturology Association norms). In the unipedal stance condition, participants stood on their dominant leg used to kick a ball placed on the floor in front of them. Participants stood on the platform in two different surface conditions: firm surface (the rigid surface of the force platform) and foam surface (surface consisted of a foam block [466 mm length × 467 mm width × 134 mm height above ground] with a density of 21.3 kg/m^3^ and an elastic modulus of 20.900 N/m^2^ mounted on the rigid surface of the force platform). The CoP sways of each participant were recorded in four conditions once they reached a steady state on the platform: firm bipedal (OEs), firm bipedal (CEs), firm unipedal, and foam bipedal. Each trial lasted 25.6 s. To avoid potential effects of fatigue and learning, a 30 s rest period was provided between trials, during which participants were allowed to sit down if needed before returning to the platform (with the evaluator verifying their correct position). The parameter that is often used in static postural evaluations is the mean velocity of the CoP (CoP_Vm_) [35].

#### 2.3.2. Dynamic Balance

Dynamic balance was evaluated by the Y Balance Test (YBT) and the Expanded Timed Get-up-and-Go test (ETUGT). The YBT is a simple clinical tool used to provide quantifiable measures of dynamic postural balance in different populations, including individuals with IDs [36,37,38]. The YBT requires the participant to balance on one leg, placing their right foot at the center of the grid, which represents their preferred kicking leg, while simultaneously reaching as far as possible with the other leg in three separate directions: anterior (ANT), posterolateral (PLAT), and posteromedial (PMED) [39]. These orientations were spaced evenly at an angle of 135 degrees from each other. To account for the influence of foot length on the test results and ensure standardized measurements, the length of each participant’s foot was assessed. This measurement was taken from the upper anterior iliac spine to the inner ankle while the participant was lying in a supine position on the ground, following established guidelines. To familiarize participants with the test procedure and optimize their performance, each participant underwent six practice trials [38]. This allowed them to gain a better understanding of how to execute the test and familiarize themselves with the required movements and balance adjustments. Reach distances were recorded by the tester placing a mark on the tape measure that recorded the touchdown point. The YBT composite score is measured by taking the sum of the three reach distances, normalizing the results to the lower limb length (anterior superior iliac spine to medial malleolus) by dividing the distance by the leg’s length (distance/length of the legs) * 100. Three trials for each test condition were performed with a 2–3 min resting period permitted between all trials. The greatest successful reach for each direction was used for analysis.

The ETUGT is a functional assessment tool that evaluates mobility and dynamic balance in individuals with IDs [10,40]. This test was conducted by a trained examiner who timed the performance with a chronometer. Participants were instructed to sit straight in a chair with their backs touching the back of the chair. The test began with verbal “go” instructions from the examiner, and the time taken to complete the test was recorded. Participants then arose from the chair, walked at their fastest walking speed (without running), turned around right or left after passing a tape (cross-market at 10 m distance from the start point) at the end of the way, returned to the start chair, turned around, and sat down. Three recorded efforts were made with 1 min rest intervals. The fastest attempt was recorded and used in the analysis [41].

### 2.4. Verbal Encouragement

During the static balance tests, participants received VE periodically throughout each trial. VE was administered approximately every 5 s during the 25.6 s trial to maintain stability. Encouraging phrases such as “You’re doing great! Keep stable until the signal, don’t move” were used to support participants in maintaining their balance. For the dynamic postural balance assessment using the YBT, participants were motivated with expressions emphasizing their abilities and encouraging them to push their limits, such as “Go, go, go!” and “Go as far as you can!” These expressions aimed to inspire participants to extend their reach and achieve their maximum potential. VE was initially provided once at the beginning of each YBT trial. VE was given multiple times (approximately 4–6 times) during a single YBT trial, with an increased frequency as the trial progressed to encourage participants to reach their maximum reach. Concerning the ETUGT, participants were encouraged using VE to enhance their walking performance. Phrases such as “Go, go! Walk fast! Yeah, you’re doing great!” were used to motivate participants to complete the test with speed and determination. VE was provided once at the start of each trial and was also given multiple times (approximately 4–6 times) during the walking, with an increased frequency towards the end of the test.

VE has been demonstrated to alter intrinsic motivational states, leading to increased motivation and potentially influencing maximal effort [42]. Hence, the order of the trials was not randomized. The “No VE” condition was consistently performed first, followed by the “VE” condition. This sequencing approach allowed participants to experience the enhanced motivation provided by the VE after completing the initial trial.

### 2.5. Statistical Analysis

Data were analyzed using SPSS 25.0 (Statistical Package for the Social Sciences Inc., Chicago, IL, USA). The normality of the data distribution was confirmed using the Shapiro–Wilk test.

For the static postural balance, three-way analyses of variance (ANOVA) with repeated measures (verbal encouragement [VE] × Group × Vision), (VE × Group × Posture), (VE × Group × Surface) were used to determine the effect of VE (no verbal encouragement [NO/VE]/VE) and/or Group (mild/moderate) and/or Vision (open eyes [OEs]/closed eyes [CEs]) and/or Posture (bipedal/unipedal) and/or Surface (firm/foam) on the CoP_Vm_ values.

Concerning the dynamic postural balance, a two-way (VE × Group) ANOVA with repeated measures was used to determine the effect of VE (NO/VE/VE) and/or Group (mild/moderate) on the Y Balance Test (YBT) scores in the anterior (ANT), posteromedial (PMED), and posterolateral (PLAT) directions, as well as on the YBT composite scores, and on the Expanded Timed Get-up-and-Go test (ETUGT) performance.

To determine whether the statistically significant differences found were practically significant, the effect size of each outcome measure was calculated. The partial eta squared (ηp^2^) formula was calculated for the main effects and interactions (small: 0.01 < ηp^2^ < 0.06; moderate: 0.06 < ηp^2^ < 0.14; large: ηp^2^ > 0.14), and Cohen’s d was calculated for the pairwise differences (trivial: d < 0.2; small: 0.2 ≤ d < 0.5; moderate: 0.5 ≤ d < 0.8; large: d ≥ 0.8) [37]. To account for multiple comparisons, a Bonferroni adjustment was conducted. The level of statistical significance was set at *p* < 0.05. A 95% confidence interval (CI) was calculated for means.

## 3. Results

### 3.1. Static Balance: CoP Values Results

In the three-way ANOVA (VE × Group × Vision), significant main effects were found for VE (F(1,23) = 28.00, *p* < 0.001, ηp^2^ = 0.54), Group (F (1,23) = 18.75, *p* < 0.001, ηp^2^ = 0.44), and Vision (F(1,23) = 24.42, *p* < 0.001, ηp^2^ = 0.51), and significant VE × Group (F(1,23) = 4.34, *p* = 0.04, ηp^2^ = 0.15) and VE × Vision × Group (F(1,23) = 8.5, *p* = 0.008, ηp^2^ = 0.28) interactions were found. However, no significant interactions were observed for VE × Vision and Group × Vision. Concerning the analysis of the three-way ANOVA (VE × Group × Posture), significant main effects only for Group (F(1,23) = 11.80, *p* = 0.002, ηp^2^ = 0.33) and Posture (F(1,23) = 223.68, *p* < 0.001, ηp^2^ = 0.91) and significant Group × Posture (F(1,23) = 5.42, *p* = 0.02, ηp^2^ = 0.19) interaction were found. However, there were no significant interactions for VE × Group, VE × Posture, or VE × Group × Posture. In the analysis of the three-way ANOVA (VE × Group × Surface), significant main effects were observed only for Group (F(1,23) = 7.79, *p* = 0.01, ηp^2^ = 0.25) and Surface (F(1,23) = 88.28, *p* < 0.001, ηp^2^ = 0.79). However, no significant interactions were found for VE × Group, Group × Surface, VE × Surface, or VE × Group × Surface.

Regarding the effect of VE, the results showed that VE significantly decreased postural sway in individuals with mild IDs during the firm bipedal CEs condition. Specifically, the post hoc analysis revealed a significant decrease (*p* < 0.001, d = 0.80) in the CoP_Vm_ values in the group with mild IDs only during the firm bipedal CEs condition when VE was provided compared to the NO/VE condition (Table 2, Figure 1).

Analyzing the effect of Group, the results showed that the group with mild IDs had significantly lower postural sway in various conditions than the moderate group. The post hoc analysis indicated that the CoP_Vm_ values in the group with mild IDs were significantly lower compared to the moderate group in the following conditions: firm bipedal stance under OEs and CEs conditions during both VE and NO/VE (NO/VE, OEs: *p* = 0.003, d = 1.30; VE, OEs: *p* = 0.002, d = 1.39; NO/VE, CEs: *p* = 0.005, d = 1.24; VE, CEs: *p* < 0.001, d = 2.03), and firm unipedal stance during both VE and NO/VE (NO/VE: *p* = 0.011, d = 1.10; VE: *p* = 0.005, d = 1.24) (Table 2, Figure 1). In addition, in the foam bipedal condition, the CoP_Vm_ values were significantly lower (*p* = 0.03, d = 0.90) in the mild IDs group than in the moderate group only during VE (Table 2, Figure 1).

Regarding the Vision factor, in the firm bipedal stance, switching from OEs to CEs resulted in a significant increase in postural sway in the moderate IDs group regardless of VE. This was indicated by significantly higher (*p* < 0.001) CoP_Vm_ values (Table 2, Figure 1). In the mild IDs group, this switch significantly increased (*p* < 0.001) the CoP_Vm_ values only during the NO/VE condition (Table 2, Figure 1).

Regarding the Posture and Surface factors, the results showed that postural sway significantly increased in more challenging conditions, regardless of the presence of VE. The post hoc analysis revealed that the CoP_Vm_ values significantly increased (*p* < 0.001) in the firm unipedal and foam bipedal conditions compared to the firm bipedal condition during both NO/VE and VE conditions.

### 3.2. Dynamic Balance

#### 3.2.1. Y Balance Test Results

There were no significant main effects of VE, Group, or VE × Group interaction in the ANT direction. Regarding the PMED and PLAT directions, significant main effects of VE were found (F(1,23) = 23.29, *p* < 0.001, ηp^2^ = 0.50; F(1,23) = 9.85, *p* = 0.005, ηp^2^ = 0.30, respectively) and Group (F(1,23) = 6.46, *p* = 0.01, ηp^2^ = 0.21; F(1,23) = 6.93, *p* = 0.01, ηp^2^ = 0.21, respectively). VE × Group interaction was observed only in PMED direction (F(1,23) = 10.70, *p* = 0.003, ηp^2^ = 0.31. In the analysis of the composite score, both VE (F(1,23) = 33.58, *p* < 0.001, ηp^2^ = 0.59) and Group (F(1,23) = 5.99, *p* = 0.02, ηp^2^ = 0.20) showed significant main effects. In adition, the VE × Group interaction was significant (F(1,23) = 14.80, *p* = 0.001, ηp^2^ = 0.39).

Regarding the effect of VE, the results showed that the VE improved YBT scores only for the group with mild IDs compared to NO/VE. Specifically, the Bonferroni test revealed a significant increase in scores for both PMED and PLAT directions, as well as the composite scores (*p* = 0.001, d = 0.46; *p* < 0.001, d = 0.43; *p* < 0.001, d = 0.40, respectively) (Table 2, Figure 2). Analyzing the effect of Group, the results showed that the mild IDs group exhibited higher YBT scores in the PMED, PLAT, and composite scores compared to the moderate IDs group (*p* < 0.001, d = 1.20; *p* = 0.006, d = 1.14; *p* = 0.02, d = 1.07, respectively) only during the VEs condition (Table 2, Figure 2).

#### 3.2.2. Expanded Timed Up-and-Go Test Results

For the ETUGT, there were significant main effects on VE (F (1,23) = 35.75, *p* < 0.001, ηp^2^ = 0.60) and Group (F) (1,23 = 5.13, p = 0.03, ηp^2^ = 0.15). However, there was no significant VE × Group interaction.

Regarding the effect of VE, the results showed that the VE significantly improved performance in ETUGT for both mild (*p* < 0.001, d = 0.80) and moderate (*p* < 0.001, d = 0.41) groups (Table 2, Figure 3). Concerning the effect of Group, the results showed that the mild IDs group had significantly higher performance in ETUGT (*p* = 0.04, d = 0.99) compared to the moderate group only during the VE condition (Table 2, Figure 3).

## 4. Discussion

### 4.1. Effect of VE

#### 4.1.1. Static Balance

The results of this study showed that the CoP_Vm_ values significantly decreased in the group of mild IDs during the firm bipedal CEs condition when VE was provided compared to the NO/VE condition. In fact, postural control involves the integration of sensory information from multiple systems, including vision, proprioception, and the vestibular system [43,44]. Individuals with IDs may experience deficits in one or more of these sensory systems, which can compromise their ability to maintain postural stability [45]. It seems that VE helps compensate for these sensory deficits and provides additional feedback and guidance to enhance postural control [28]. During the firm bipedal CEs condition, individuals with mild IDs may rely predominantly on proprioceptive and vestibular inputs to maintain balance, as visual information is removed [46]. This condition places higher demands on internal body awareness and proprioceptive processing [46]. The significant decrease in CoP_Vm_ values observed during this condition suggested that VE may improve the participants’ ability to utilize proprioceptive and vestibular cues effectively, leading to enhanced static postural balance. The positive effect of VE on postural balance in the group of mild IDs highlighted the importance of providing external cues and support to individuals with IDs during balance tasks.

It is important to note that the lack of significant effects of VE on CoP_Vm_ values in other postural conditions, such as firm bipedal OE, firm unipedal, and foam bipedal for both groups, suggested that the effectiveness of VE may vary depending on the specific postural task and sensory demands involved. It seems that factors such as task complexity, sensory integration requirements, and individual characteristics may influence the impact of VE on postural control. Our study highlights the need for future studies to delve deeper into these individual differences and task-specific factors to elucidate the circumstances where VE may be less effective.

#### 4.1.2. Dynamic Balance

The study results showed that VE had a positive impact on the YBT scores in PMED and PLAT, directions, and on the YBT composite scores for the group of mild IDs. Importantly, the study findings revealed that both the mild and moderate IDs groups demonstrated improved performance in the ETUGT when VE was provided compared to the NO/VE condition. This indicates that VE, in the form of verbal support and motivation, had a positive impact on the functional mobility of both groups, suggesting that individuals with mild and moderate IDs can benefit from VE as a motivational strategy to enhance their functional mobility. The study results suggested that VE has a positive impact on dynamic balance in individuals with mild IDs. Interestingly, these findings align with a previous study conducted on individuals with chronic ankle instability [19]. In that study, significant improvements were observed in the Star Excursion Balance Test scores during VE compared to the NO/VE condition in individuals with chronic ankle instability when compared to healthy individuals. This parallel suggests that the positive effects of VE on balance are not limited to individuals with IDs but can also be observed in other populations with balance-related challenges. Moreover, in a related study involving typically developing individuals, it was demonstrated that VE led to an increase in the mean peak force in the elbow flexors [24]. This suggests that VE not only impacts postural balance but also has the potential to enhance muscular strength and force generation in specific muscle groups. While the present study focused on postural balance outcomes, it is worth noting that the benefits of VE may extend beyond postural balance alone, potentially encompassing other aspects of motor performance and physical function [20,24,30,47]. Furthermore, it has been indicated that sudden loud noises can improve motor performance by reducing supraspinal inhibition [24]. In addition, VE has been shown to increase focus on motor tasks and activate higher brain centers [27]. Based on the results of the current study, it seems that VE may help individuals with IDs increase their confidence and focus in performing dynamic balance tasks [19,27]. This finding highlighted the importance of psychological factors in balance tasks and emphasizes the role of VE in fostering a sense of confidence [27,30,31] and possibly in improving dynamic postural balance. While the current study revealed notable enhancements in YBT scores for the PMED and PLAT directions, as well as the composite scores, it is important to address the absence of such improvements in the ANT direction. This discrepancy implies that the ANT direction may pose unique challenges related to coordination, muscle engagement, and confidence levels among participants [48]. The act of reaching forward while simultaneously maintaining balance on one foot in the ANT direction could be inherently demanding, which might explain the lack of significant effects of VE alone. Further research should explore interventions to better support individuals with IDs in improving their performance in the challenging ANT direction of the YBT.

### 4.2. Effect of Group

The results of our study indicate that individuals with mild IDs demonstrated better static postural balance than those with moderate IDs in specific conditions. The CoP_Vm_ values were greater in the group of mild IDs than in the moderate group in the firm bipedal stance under OEs and CEs, in the firm unipedal stance during both VE and NO/VE, and in the foam bipedal stance during VE. These findings suggest that the severity of IDs plays a significant role in determining the extent of motor impairments, with individuals with moderate IDs experiencing more pronounced deficits in motor skills than those with mild IDs. A previous study focused on understanding the effects of IDs on motor performance, further supporting our findings [49]. The findings of that study revealed that individuals with moderate IDs tend to exhibit greater motor skill impairments than those with mild IDs. This highlights the crucial role that the severity of ID plays in determining motor performance capabilities, such as postural balance. It has been suggested that the level of IQ and cognitive function can impact motor skills [50,51]. Similarly, previous studies have shown that lower IQ scores are associated with poor postural stability, particularly in children with autism [52,53]. While these findings offer strong explanations for our own results, highlighting the connection between the different levels of ID and static postural balance, further studies are still necessary to better understand this relationship.

### 4.3. Effect of Vision

The results showed that the switch from OEs to CEs in a bipedal stance significantly increased the CoP_Vm_ values during the NO/VE and VE conditions in the group with moderate IDs and only during the NO/VE condition in the group with mild IDs. The observed increase in CoP_Vm_ values during CEs in the NO/VE condition suggested that the absence of visual feedback disrupted the balance control mechanisms for individuals with mild and moderate IDs, resulting in compromised static postural balance. On the other hand, the absence of a significant increase in CoP_Vm_ values during CEs in individuals with mild IDs during the VE condition indicated that VE may have a compensatory effect. In fact, VE has been shown to enhance motivation, focus, and attention, which can positively influence motor performance [19,27,30,31]. In this context, in the presence of VE, individuals with mild IDs may be more confident in their ability to maintain their static postural balance even without visual cues. It seems that the VE acts as an external cue, guiding their attention and efforts toward the task at hand, potentially minimizing the destabilizing effect of CEs in the group with mild IDs. This finding suggested that the motivational aspect of VE may play a role in improving static postural balance during challenging sensory conditions.

### 4.4. Effect of Posture and Surface

The results regarding the posture and surface factors indicate that there were significant increases in the CoP_Vm_ values when transitioning from the firm bipedal condition to both the unipedal and foam bipedal conditions. This result was consistent across both the NO/VE and VE conditions. The increase in CoP_Vm_ values suggested that the transition from the stable firm bipedal stance to the more challenging unipedal and foam bipedal stances resulted in greater postural sway and instability. This finding indicates that individuals experienced more difficulty maintaining their balance in these challenging postural conditions compared to the relatively easier task of maintaining balance on both legs in the firm bipedal condition [54]. The results emphasize the important role of postural and surface conditions in influencing static postural balance in this population. Indeed, when individuals shift from a stable bipedal stance to a unipedal stance or to standing on a foam surface, the demands on their postural control system increase significantly [54,55]. Furthermore, maintaining postural balance on one leg or on an unstable foam surface requires greater motor control, coordination, and sensory integration compared to the more stable bipedal condition, resulting in greater postural sway and instability [54,56].

### 4.5. Limitations

This study faces some limitations that should be addressed in future studies. Our investigation focused solely on the immediate effects of VE on static and dynamic postural balance. We did not investigate the long-term effects or the sustainability of any improvements observed. Therefore, the generalizability of our findings to long-term interventions may be limited. Future studies should include follow-up assessments conducted over an extended period to determine if the benefits of VE endure over time. This would provide a better comprehension of the potential long-term effectiveness of VE interventions for improving postural balance in individuals with IDs. In addition, our study did not explicitly investigate factors such as mood, self-confidence, motivation, and attentional focus, which may have an impact on the effectiveness of VE. Examining these mechanisms could give better explanations of our findings. Therefore, we recommend that future studies incorporate measures and assessments to capture these variables, facilitating a more comprehensive understanding of the psychological and physiological mechanisms involved.

### 4.6. Practical Implications

VE proves to be a motivational strategy in balance tasks in individuals with IDs, particularly those with mild IDs. Therapists and practitioners should consider using positive and supportive verbal cues to enhance motor performance and engagement in balance tasks to prevent falls. By incorporating VE during assessments, the focus and determination of participants can significantly improve, enhancing motor performance and engagement during balance tasks, as well as helping in fall prevention, a crucial consideration for individuals with IDs. This improvement is a direct enhancement of their overall well-being and capabilities, allowing them to lead more independent lives. It is important to note that assessing postural balance with and without VE is crucial in understanding the abilities of individuals with IDs. While testing without external cues like VE provides a baseline assessment, reflecting intrinsic capabilities unaffected by external stimuli, real-world scenarios often involve verbal cues or support during therapy or daily activities. Therefore, incorporating standardized VE in assessments is equally important. This dual approach allows for a comprehensive evaluation, providing a better understanding of both intrinsic abilities and performance under supportive conditions. By considering both perspectives, researchers can better understand the nuanced aspects of postural control abilities in individuals with IDs. Moreover, although testing provides valuable information about baseline abilities and immediate responses to VE, its sustained application during regular training and physical therapy can have a more profound impact on long-term skill development. Consistent VE and support provided during therapeutic exercises and activities have the potential to enhance movement capabilities, improve self-confidence, and promote motor learning over time. Therefore, by integrating standardized VE into training and therapeutic interventions, individuals with IDs can experience greater engagement, motivation, and progress in improving their postural control abilities, offering significant potential for lasting enhancements in functional abilities and overall quality of life.

## 5. Conclusions

This study concluded that VE has positive effects on static balance under the firm bipedal CEs condition and dynamic balance in individuals with IDs. The findings suggested that VE can improve postural balance and functional mobility, as evidenced by improved performance in the YBT and ETUGT. These results highlighted the importance of incorporating VE as a motivational strategy in the balance tasks for individuals with IDs, particularly those with mild IDs. Therapists and practitioners should consider using positive and supportive verbal cues to enhance motor performance and engagement in balance tasks to prevent falls. Future research should further explore the underlying mechanisms and long-term effects of VE, as well as its impact on psychological and physiological outcomes.

## Figures and Tables

**Figure 1 healthcare-12-00995-f001:**
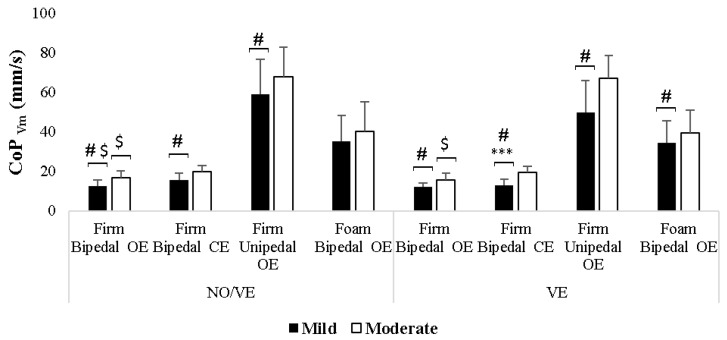
Comparison of CoP_Vm_ values between adults with mild and moderate intellectual disabilities (IDs) in bipedal stance, in open eyes (OEs) and closed eyes (CEs) conditions, and unipedal/foam stances in OEs condition during no verbal encouragement (NO/VE) and verbal encouragement (VE) conditions. Notes: ***: significant difference (*p* < 0.001) between NO/VE and VE; #: significant difference (*p* < 0.05) between mild IDs and moderate IDs groups; $: significant difference (*p* < 0.001) between OEs and CEs conditions.

**Figure 2 healthcare-12-00995-f002:**
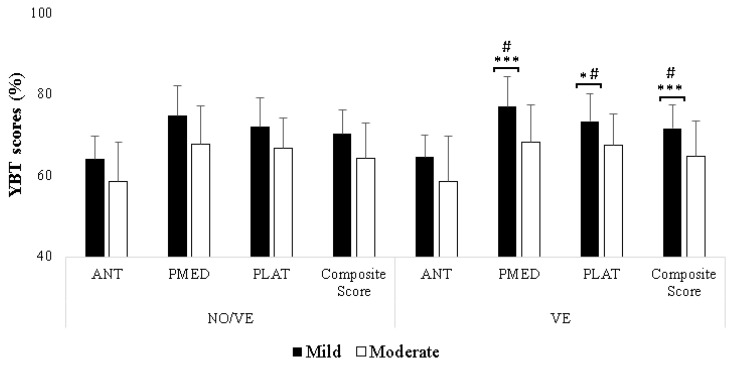
Comparison of Y Balance Test (YBT) composite score and scores in anterior (ANT), posteromedial (PMED), and posterolateral (PLAT) for adults with mild and moderate intellectual disabilities (IDs) during no verbal encouragement (NO/VE) and verbal encouragement (VE) conditions. Notes: *, ***: significant differences (*p* < 0.05, *p* < 0.001) between NO/VE and VE; #: significant difference (*p* < 0.05) between mild IDs and moderate IDs groups.

**Figure 3 healthcare-12-00995-f003:**
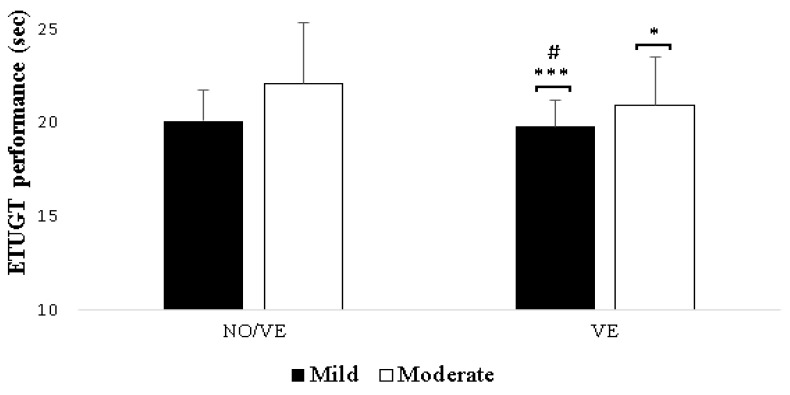
Comparison of Expanded Timed Up-and-Go Test (ETUGT) performance between adults with mild and moderate intellectual disabilities (IDs) during no verbal encouragement (NO/VE) and verbal encouragement (VE) conditions. Notes: *, ***: significant differences (*p* < 0.05, *p* < 0.001) between NO/VE and VE; #: significant difference (*p* < 0.05) between mild IDs and moderate IDs groups.

**Table 1 healthcare-12-00995-t001:** Participants’ characteristics.

	Mild (N = 13) m ± (SD)	Moderate (N = 12) m ± (SD)	Degree of Freedom	Independent *t*-Test
Gender	Male	Male	-	-
Age (years)	27.30 ± (3.30)	26.83 ± (4.72)	23	*p* = 0.77
Height (cm)	168.46 ± (7.05)	165.70 ± (7.12)	23	*p* = 0.34
Weight (kg)	64.10 ± (6.38)	67.69 ± (6.42)	23	*p* = 0.17
IQ	61.92 ± (4.66)	46.25 ± (3.67)	23	*p* < 0.001
Living status	Middle Socio-economic	Middle Socio-economic	-	-

**Table 2 healthcare-12-00995-t002:** Means (SD) of the static and dynamic postural balance during no verbal encouragement (NO/VE) and verbal encouragement (VE) interventions in adults with mild intellectual disabilities (IDs) compared to adults with moderate IDs.

	Mild		Moderate	
	Means (SD)	95% CI	Means (SD)	95% CI
Static postural balance				
Firm Bipedal OEs NO/VE	12.76 (2.88)	11.02 to 14.50	16.85 (3.37)	14.70 to 18.99
Firm Bipedal OEs VE	12.11 (2.07)	10.86 to 13.36	15.84 (3.18)	13.82 to 17.86
Firm Bipedal CEs NO/VE	15.78 (3.40)	13.72 to 17.84	19.94 (3.28)	17.85 to 22.03
Firm Bipedal CEs VE	13.19 (3.08)	11.33 to 15.06	19.54 (3.15)	17.53 to 21.55
Firm Unipedal NO/VE	51.07 (16.18)	41.29 to 60.86	68.25 (14.84)	58.22 to 77.69
Firm Unipedal VE	49.89 (15.90)	40.28 to 59.50	67.22 (11.69)	59.79 to 74.65
Foam Bipedal NO/VE	30.52 (9.61)	24.71 to 36.33	40.51 (14.77)	31.12 to 49.90
Foam Bipedal VE	29.92 (9.00)	24.48 to 35.36	39.41 (11.92)	31.84 to 46.99
Dynamic postural balance				
YBT ANT NO/VE	63.99 (5.60)	60.60 to 67.38	58.43 (9.79)	52.21 to 64.65
YBT ANT VE	64.56 (5.47)	61.25 to 67.87	58.53 (11.07)	51.49 to 65.56
YBT PMED NO/VE	74.70 (7.43)	70.20 to 79.19	67.75 (9.38)	61.79 to 73.71
YBT PMED VE	78.06 (7.34)	73.62 to 82.50	68.40 (8.68)	62.88 to 73.92
YBT PLAT NO/VE	72.70 (5.49)	69.38 to 76.02	66.83 (7.20)	62.25 to 71.41
YBT PLAT VE	75.02 (5.37)	71.77 to 78.26	67.34 (7.83)	62.37 to 72.32
YBT Composite score NO/VE	70.46 (5. 47)	67.15 to 73.77	64.34 (8.51)	58.92 to 69.75
YBT Composite score VE	72.54 (5.26)	69.36 to 75.73	64.76 (8.73)	59.21 to 70.31
ETUGT NO/VE	20.09 (1.67)	19.08 to 21.11	22.10 (3.23)	20.05 to 24.16
ETUGT VE	18.79 (1.40)	17.93 to 19.64	20.89 (2.64)	19.21 to 22.57

Notes: VE: verbal encouragement; NO/VE: no verbal encouragement; OEs: open eyes; CEs: closed eyes; YBT, Y Balance Test; ANT, anterior; PMED, posteromedial; PLAT, posterolateral; ETUGT, Expanded Timed Up-and-Go Test.

## Data Availability

The data that support the findings of this study are available on request from the corresponding author. The data are not publicly available due to privacy or ethical restrictions.

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
