# Peer review of "Effect of Verbal Encouragement on Postural Balance in Individuals with Intellectual Disabilities"

_healthcare, 2024, doi:10.3390/healthcare12100995_

Round 1

Reviewer 1 Report

Comments and Suggestions for Authors

The article needs to be corrected.

Author Response

Dear Reviewer 1

Thank you for  your time and advice

Reviewer 2 Report

Comments and Suggestions for Authors This study appears to be of some value as it investigate the effects of vocal encouragement on static and dynamic postural retention in persons with mild and moderate intellectual disabilities.
The statistical analysis methods used in this study appear to be appropriate, revealing some significant differences in the results based on the degree of disability, the content of the test, and whether or not the participants were encouraged with or without vocal encouragement.
While these results are worth presenting, the fact that vocal encouragement contributes to improved performance per se is already well known in the field of sports training and physical therapy.
Of interest to readers of this paper is that if vocal encouragement has a substantial effect on the outcome of postural control, should we test intellectual disabilities under conditions without external interference such as vocal encouragement in order to properly assess their postural control abilities, or, conversely, should we test them under conditions that maximize their performance? Should standardized vocal encouragement be used to maximize performance, or can the results of both provide useful information on the evaluation of ability and the likelihood of improvement through training? The second question is whether vocal encouragement during regular training and physical therapy, rather than testing, can result in long-term improvement in ability.
If authors could discuss these two points in the discussion while providing evidence, it would contribute to making this paper more valuable.

Author Response

Dear Reviewer 2,

Thank you for your time and your advice.

Reviewer 3 Report

Comments and Suggestions for Authors At the end of the introduction, clearly present the research hypothesis you want to conduct in this study   193 line 2.3.1. Dynamic balance  - > 2.3.2 339 line 3.2.1. Expanded Timed Up and Go Test results -> 3.2.2   470 Limitations -> 4.5 practical implications next 4.6 Limiations modify or 4.5 Limitations 4.6 Practical implications modify

I think a meaningful study has been conducted to find out the effect of VE on ID. Thank you for giving me the opportunity to review enjoyable research topics. I think the completion is high enough to be published even in the current state. Please review it carefully until the end to increase the completion.

Author Response

Dear Revuewer 4

Thank you for your time and your advice.

Round 2

Reviewer 1 Report

Comments and Suggestions for Authors

Thank you for incorporating the amendments the article looks much better and more interesting for the reader.